# Ecological Strategy Spectra for Communities of Different Successional Stages in the Tropical Lowland Rainforest of Hainan Island

Chen Chen [1], Yabo Wen [1], Tengyue Ji [1], Hongxia Zhao [1], Runguo Zang [2,3] and Xinghui Lu [1,*]

[1] College of Agronomy and Agricultural Engineering, Liaocheng University, Liaocheng 252000, China; cc2842855056@163.com (C.C.); wenyabo416@163.com (Y.W.); jity923@163.com (T.J.); zhaohxia1314@163.com (H.Z.)

[2] Key Laboratory of Forest Ecology and Environment of the National Forestry and Grassland Administration, Research Institute of Forest Ecology, Environment, and Protection, Chinese Academy of Forestry, Beijing 100091, China; zangrung@caf.ac.cn

[3] Co-Innovation Center for Sustainable Forestry in Southern China, Nanjing Forestry University, Nanjing 210037, China

[*] Correspondence: luxinghui_0@163.com

**Abstract:** Plant ecological strategies are shaped by long-term adaptation to the environment and are beneficial to plant survival and reproduction. Research is ongoing to better understand how plants best allocate resources for growth, survival and reproduction, as well as how ecological strategies may shift in plant communities over the course of succession. In this study, 12 forest dynamics plots in three different successional stages were selected for study in the tropical lowland rainforest ecosystem of Hainan Island. For each plot, using Grime's competitor, a stress-tolerator, the ruderal (CSR) scheme and using the CSR ratio tool "StrateFy", an ecological strategy spectrum was constructed using functional trait data obtained by collecting leaf samples from all woody species. The ecological strategy spectra were compared across successional stages to reveal successional dynamics. The results showed: (1) The ecological strategy spectra varied among forest communities belonging to three different successional stages. (2) The community-weighted mean CSR (CWM-CSR) strategies shifted with succession: CWM-S values decreased, while the CWM-C and CWM-R values increased. Overall, shifts in plant functional traits occurred slowly and steadily with succession showing complex and diverse trade-offs and leading to variation among the ecological strategy spectra of different successional stages.

**Keywords:** ecological strategy; succession; functional traits; forest vegetation; successional dynamics; ecosystem function

## 1. Introduction

Ecological strategies reflect how species optimally allocate resources to growth, survival and reproduction, thereby, capturing trade-offs among functional traits [1,2]. After selection by abiotic and biotic environmental factors, a plant's ecological strategy represents its ideal combination of traits [3,4]. To remain competitive within their ecological community, plants may adjust their resource allocations. Species display different combinations of traits [5,6] based on their tolerance of current environmental conditions and ability to cope with resource-poor habitats [7,8].

The study of ecological strategies is an important avenue to understanding biological community assembly and dynamics in response to environmental change [9]. Many famous ecologists have investigated different aspects of species' ecological strategies [10,11]. Additionally, the study of plant functional traits to better our understanding of plant ecological strategies represents a current research hotspot in ecology [12–14].



Grime's competitor, stress-tolerator, ruderal CSR theory is foundational to research on ecological strategies and is also based on plant functional traits [15,16]. Recent developments in CSR theory seek to explain plant ecological strategies in terms of the primary dimension of functional trait variation. Pierce et al. [1] described how three functional traits (of plant leaves) can be used to extrapolate the three main dimensions of functional trait variation and developed a freely available tool, "StrateFy", to implement these calculations.

Plant species are divided into 19 strategy types, including three primary strategies (C, S and R), four secondary strategies (CS, CR, SR and CSR) and twelve tertiary strategies (C/CR, C/CS, C/CSR, CR/CSR, CS/CSR, CS/CSR, R/CR, S/CS, R/CSR, S/CSR, SR/CSR, R/SR and S/SR) [17,18]. Secondary strategy and tertiary strategy are different combinations of the three primary strategies. The three primary strategies (i.e., C, S and R) represent combinations of traits best suited to competition (C), abiotic stress tolerance (S) and ruderal habitats with periodic biomass destruction (R) [1]. A more detailed description refers to CSR classification after Hodgson et al.'s CSR classification [19].

Using CSR theory and community-weighted mean (CWM) trait values, differences in the functional compositions among plant communities can be evaluated. Community-level ecological strategy spectra can then be estimated using the CWM trait values in "StrateFy". These ecological strategy spectra (i.e., the number and relative abundance of species holding different ecological strategy types in the community) can provide a "functional summary" of the vegetation, which can also be used to study how communities vary among successional stages [17].

Grime's CSR theory has been applied to functional analyses, involving global [20], regional [21] and local [22] scales. Previous studies have shown that ecological strategies can be used to explain the distribution of species along environmental gradients [23,24]. Shifts in ecological strategy can reflect the influence of environmental gradients and disturbances of forest dynamics [25]. While there have been many studies of vegetation function and community assembly [7,26], little is known about how community ecological strategy spectra shift with succession.

Succession represents a process of dynamic community construction [27]. Across successional stages, the environmental factors affecting the vegetation are constantly changing [28], as are plant–environment interactions. As a result, the community composition and structure shift over time [29], thus, affecting ecosystem function [30]. Although there are still disagreements about the predictability of community structure and the role of historical contingency [31], many studies have shown that, while species composition during succession is often unpredictable, functional changes are deterministic [32].

Functional analyses of communities may be helpful to better characterize the successional process and related environmental changes [33]. Studying ecological strategies at the community-level may provide insights into the resource balance at each restoration stage, as well as the process of community assembly and ecosystem functioning [34]. However, the knowledge of how ecological strategy spectra of different successional forests in the same region in tropical forests is largely unclear.

In restoring abandoned slash-and-burn farmland in the Bawangling tropical lowland rainforest on Hainan Island, China, communities of different restoration ages have been produced. This region therefore provides an ideal system to evaluate the relationship between restoration age and the ecological strategy spectrum. In this study, forests ecological strategy spectra belonging to different successional stages were determined. Then, the following two questions were discussed: (1) Do the forest ecological strategy spectra change with succession? (2) What is the effect of succession on forest ecological strategy composition?

This approach is valuable for expanding the study of successional forest ecosystem functioning [35]. Succession is hypothesized to affect a plant community's ecological strategy spectrum, with the community-weighted mean S expected to be dominant in the later stages of succession and community-weighted mean C dominant early on. Moreover, plant ecological strategies may become more specialized over the course of succession.

## 2. Materials and Methods

### 2.1. Study Area and Sampling Strategy

This study was carried out in the Bawangling Forest Region on Hainan Island (18°52′–19°12′ N, 108°53′–109°20′ E), which occurs at the northern limit of the tropical rainforest in Asia [36]. Bawangling Forest covers an area of about 500 km², with an altitudinal range from 100–1654 m. Vegetation in the area varies with altitude; however, this study focused on the tropical lowland rainforest (<800 m above sea level).

Abandoned slash-and-burn farmland, naturally-restored secondary forest and a few undisturbed old-growth forests are distributed here. Information on the history of land-use for the plots was obtained from the management records of the Bawangling National Nature Reserve [37]. Within the region, the annual average temperature is 23.6 °C, and the annual precipitation is 1677 mm. The rainy season occurs from May to October, and the dry season from November to April [36].

According to best practices published by the Center for Tropical Forestry Science (CTFS) [38], twelve forest dynamic monitoring plots of 100 × 100 m were established (twelve plots belong to three successional stages: 30-year-old secondary growth forest, 60-year-old secondary growth forest and the old growth forest) (Table 1). For the convenience of community survey, each plot was divided into 25 quadrats of 20 × 20 m, and cement piles were used to mark the four corners of each quadrat. In these fixed plots, all woody stems with diameter breast height (DBH) > 1 cm were surveyed, and the species name and DBH were recorded.

**Table 1.** Overview of forest dynamic monitoring plots in communities of various successional stages.

| Stages of Succession | Abbreviation | Interference History | Number of Plots |
|---|---|---|---|
| Early succession | E | 30-year-old secondary forest | 4 |
| Mid-succession | M | 60-year-old secondary forest | 4 |
| Late succession | O | Old growth forest | 4 |

### 2.2. Determination of Functional Traits

Ten individuals were sampled from each species (excluding endangered species) in each plot. Random sampling was conducted if there were more than ten individuals in each species. In cases where there were less than ten individual per species, all individuals were sampled. Five to ten mature leaves were collected to measure the leaf functional traits in the field. To ensure that the leaf materials remained fresh, samples were stored in fresh-keeping bags and transported to the laboratory for measurement within 24 h [39]. Two healthy leaves were selected for each individual. These were weighed (to determine the leaf fresh weight [LFW], in mg), then scanned on a flatbed scanner, and the area of each leaf (or leaf area [LA]) was determined using ImageJ. Afterwards, leaf samples were dried in an 80 °C oven to a constant weight. The leaf dry weight (LDW, in mg) was measured, and the specific leaf area (SLA) and leaf dry matter content (LDMC) were calculated [39]. Leaf traits were measured for 434 species in total.

### 2.3. Data Analysis

Using the CSR ratio tool "StrateFy" [1], one can measure three simple and easy-to-quantify leaf functional traits (the leaf area, specific leaf area and leaf dry matter content) and then use the trade-offs among them to express the degree of C-, S- and R-selection. This enables the classification of plant species according to their CSR strategy, as well as quantitative comparisons among different species or communities [19].

According to the results of the community survey and functional traits determination, we created two matrices of twelve forest dynamic monitoring plots: one is the species abundance matrix, and the other is the three functional traits matrix of leaves (leaf area, specific leaf area and leaf dry matter content) of species. The "dbFD" function in the "FD" package [40] was used to calculate community-weighted mean (CWM) values of

leaf area, specific leaf area and leaf dry matter content [41,42]. These were then input into "StrateFy" to obtain the CSR value of each species to determine the classification of the ecological strategies.

Then, we calculated the CWM-CSR value of each plot and the number and relative abundance of species with different ecological strategies in each succession stage. For each plot, the incidence of each ecological strategy was summed over the individuals to obtain the overall "strategy richness". Similarly, the distribution of each strategy in each sample plot was compiled using a three-level classification scheme (CSR). Variation in the CWM-CSR values, strategy richness and strategy distributions among successional stages was assessed using one-way ANOVAs followed by Tukey tests [43]. To compare community types based on these ecological strategy spectra, nonparametric Kruskal–Wallis tests and Wilcoxon tests were implemented.

To further explore whether communities varied in terms of their ecological strategy composition, we established the abundance matrix of each ecological strategy in 12 sample plots. Non-metric multidimensional scaling (NMDS) was utilized. The NMDS was then constrained by community type (i.e., successional stage) to evaluate how successional stage influenced variation in plant ecological strategies. We used an analysis of similarities (ANOSIM) to test the significance of the constrained axes. The "vegan" package in R was used to perform the NMDS (with the "metaMDS" function) and the ANOSIM ("ANOSIM" function). All statistical analyses were conducted in R [44].

The C, S and R values for each species were used to create a "trade-off triangle" in order to compare among target species. For each succession stage, a ternary (or triangle) plot was drawn with each axis representing a strategy (i.e., C, S or R). Individual species were then positioned within the resulting CSR triangle. Ecological strategies are represented by color: pure red indicates the C strategy, pure green indicates the S strategy, and pure blue indicates the R strategy. We copy-and-pasted the "Color values in SigmaPlot format" in "StrateFy" into Sigmaplot to obtain the colors of species. Triangle diagrams were drawn in SigmaPlot.

## 3. Results

### 3.1. Types of Community Ecological Strategies in Successional Stages

A total of 434 plant species were identified across all sample plots. We documented 182 species in forest plots of early succession, 247 species in mid-succession and 320 species in late succession. The three successional stages differed in terms of the species diversity. In our study, species were assigned to 16 out of the total 19 ecological strategies by "StrateFy", with two of the three primary strategies being C and S. There were three secondary strategies identified, namely CR, CS and CSR, and eleven tertiary strategies identified: C/CR, C/CS, C/CSR, CR/CSR, CS/CSR, R/CR, R/CSR, S/CS, S/CSR, S/SR and SR/CSR (Table 2).

**Table 2.** The types of community-level ecological strategies identified in different successional stages.

| Early Succession | Mid-Succession | Late Succession |
|:---:|:---:|:---:|
| C | C | C |
| C/CR | C/CR | C/CR |
| C/CS | C/CS | C/CS |
| C/CSR | C/CSR | C/CSR |
| CR | CR | CR |
| CS | CR/CSR | CR/CSR |
| CS/CSR | CS | CS |
| CSR | CS/CSR | CS/CSR |
| CR/CSR | CSR | CSR |
| R/CSR | R/CR | R/CSR |
| S | S | S/CS |
| S/CS | S/CS | S/CSR |
| S/CSR | S/CSR | S |
| S/SR | SR/CSR | |
| SR/CSR | | |

(Early succession = E, mid-succession = M and late succession = O).

### 3.2. Successional Dynamics in Community Ecological Strategy Spectra

The ecological strategy spectra varied among forest communities belonging to three different successional stages. The strategy richness represents the total number of CSR strategy types identified for each successional stage. In total, 16 strategy types (of 19 possible types) were distinguished: 15 in the early successional stage community, 14 in the mid-successional stage community and 13 in the mature forest (Table 2). The strategy richness showed a downward trend with succession.

In early and mid-successional stage communities, species were largely concentrated in the CS area of the ternary plot, and the three dominant strategies were CS, S/CS and CS/CSR. In the early successional stage community, no species adopted the R/CR strategy, while no S/SR and R/CSR strategies were found in the mid-successional stage community. In the late successional stage community, species were concentrated in the CS/CSR area of the ternary plot, with CS/CSR, CS and S/CS being the most common strategies. The R/CR, S/SR and SR/CSR strategies were not identified (Figures 1 and 2).

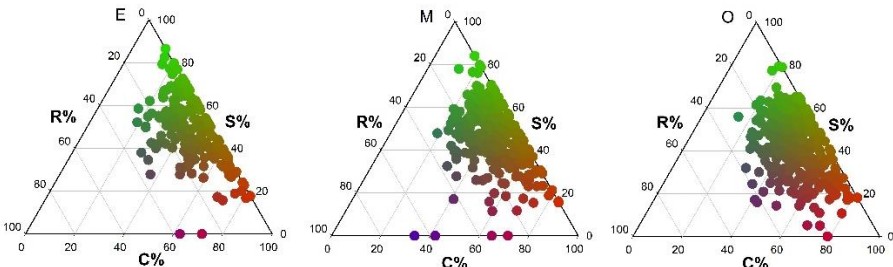

**Figure 1.** Ternary plots of species ecological strategies for different successional stages in tropical lowland rainforests on Hainan Island (E = early succession, M = mid-succession and O = late succession). C (%), S (%) and R (%) represent the three strategy components C, S and R, respectively. Ecological strategies are represented by color: pure red indicates the C strategy, pure green indicates the S strategy, and pure blue indicates the R strategy. Intermediate (mixed) colors indicate the full range of intermediate strategies (e.g., green/blue = SR strategy).

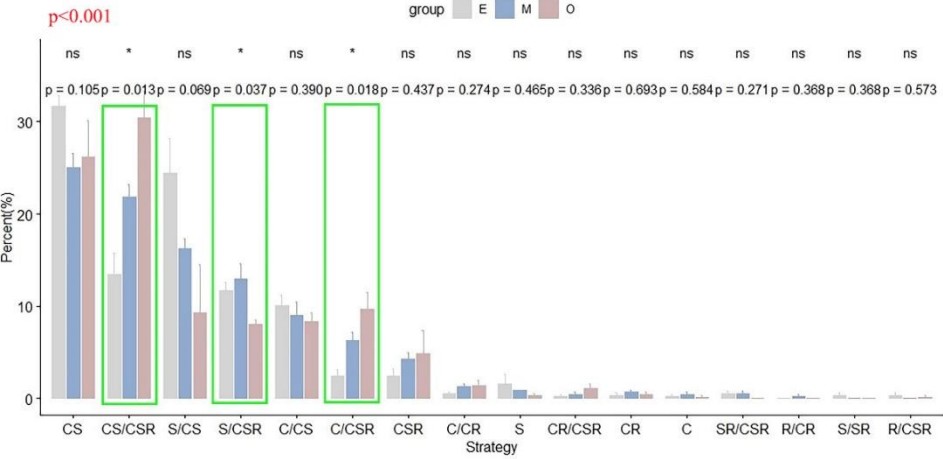

**Figure 2.** Changes in the ecological strategy spectra (based on CSR theory) across successional stages in the tropical lowland rainforest of Hainan Island (gray, E, early succession; blue, M, mid-succession; and red, O, late succession). Percentages (%) indicate the average number of species in each strategy category ($n$ = 4), with error bars denoting the standard errors. * indicates a significant difference among stages ($p < 0.05$), while ns indicates no significant difference ($p > 0.05$); green boxes additionally highlight significant cases. $p < 0.001$ in the top left indicates a significant difference among the ecological strategy spectra of three succession stages.

Based on the variance partitioning analysis, the three successional stages differed significantly in terms of the proportion of S/CSR, C/CSR and CS/CSR strategies adopted

by the species within each community (Figure 2). As succession proceeded, the proportion of S, C/CS and S/CS strategies decreased, while the proportion of C/CR, CSR, CR/CSR, C/CSR and CS/CSR strategies increased (Figure 2).

### 3.3. Effects of Successions on CSR Strategies Composition

The first NMDS axis separated the early and mid-successional stages from the late successional stage, with partial overlap between the early and middle stages (Figure 3). In addition, the ANOSIM (R = 0.6366, *p* = 0.001 < 0.05) confirmed significant variation among the ecological strategy spectra of different successional periods. The S/CSR, C/CS and CS strategies were common in all communities. However, in the NMDS, the distance between the C/CSR, CS/CSR and C/CR strategies was reduced in the late successional stage community versus the early and mid-successional stage communities, indicating that there were more species holding these strategies in mature forest (i.e., later in succession) (Figure 3).

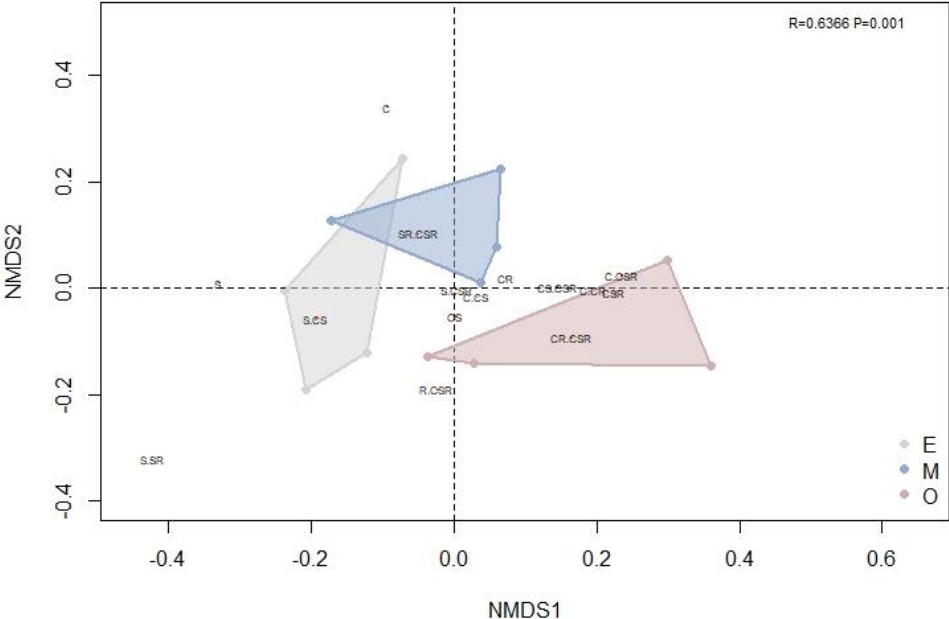

**Figure 3.** A non–metric multidimensional scaling (NMDS) diagram based on species richness was used to compare forest communities of different successional stages in the tropical lowland rainforest of Hainan Island. Polygons of different colors represent different successional stages (gray, E, early succession; blue, M, mid-succession; and red, O, late succession). Text labels within the plot represent ecological strategies (C = competitors, S = stress tolerators and R = ruderals); please refer to the CSR classification of Hodgson et al. (1999) [19] for a more detailed description. For the NMDS, R = 0.6366 and *p* = 0.001.

The average CWM-CSR values calculated for each of the four sample plots were used to represent the CSR values for each successional stage. For all three stages, the C and S components contributed more to the CWMs when compared with the R component. The CWM-C, CWM-S and CWM-R values varied among successional stages. With succession, component C increased from 38.19% to 42.22%, while component S declined from a maximum value of 53.90% to 45.35% (Figure 4). Further analysis for component C revealed a significant difference between late versus early/mid- successional stages but no difference between early and mid-successional stages. For component S, the early and mid-stages differed; however, the middle and late stages did not. Similarly, for the R component, the middle and later stages did not differ; however, both were greater than for the early successional stage (Figure 4).

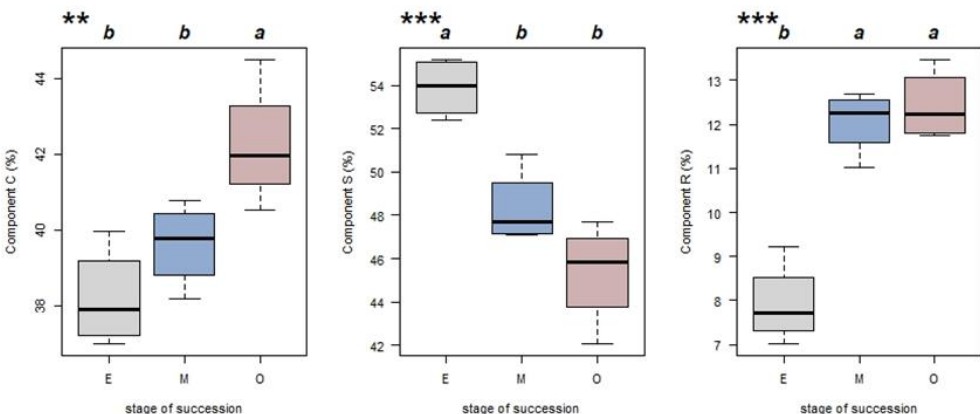

**Figure 4.** CWM-CSR values (for C, S and R strategy components) for tropical lowland rainforest stands on Hainan Island at different successional stages (gray, E, early succession; blue, M, mid-succession; and red, O, late succession). ** $p < 0.05$ and *** $p < 0.01$ indicate significant differences according to a one-way ANOVA ($n = 4$). Different letters indicate significant differences between stages (Tukey's test; $p < 0.05$).

Ecological strategies were classified into three groups: primary, secondary and tertiary. The proportion of species belonging to each group is shown in Figure 5 for each successional stage. For all three successional stages, the proportion of species increased with the group number. The results showed that most of the species in the three succession stages adopted the ecological strategy with complicated trade-offs. A few species adopted the primary strategies (the C and S ecological strategies).

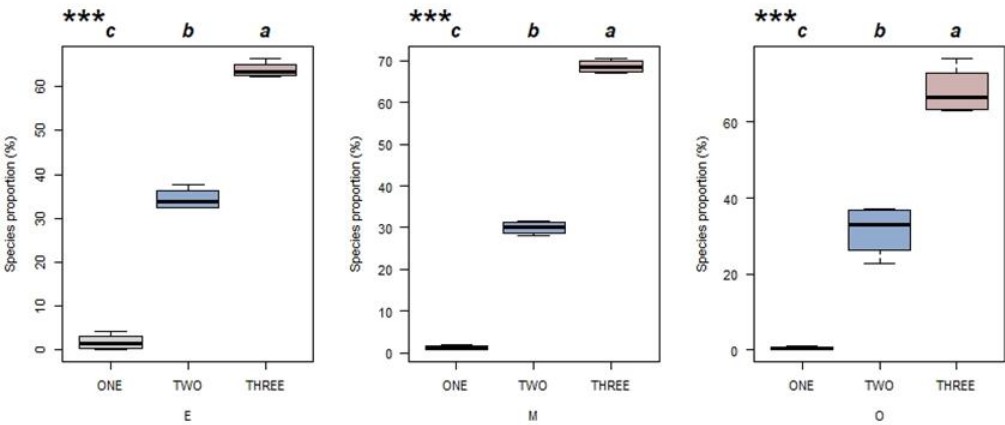

**Figure 5.** The proportion of species in Hainan tropical lowland rainforest stands belonging to each of three ecological strategies groups for multiple successional stages (E = early succession, M = mid-succession and O = late succession). Primary strategies (in grey) include C and S. Secondary strategies (in blue) include CS, CR and CSR. Tertiary strategies (in red) include C/CR, C/CS, C/CSR, CR/CSR, CS/CSR, R/CR, R/CSR, S/CS, S/CSR, S/SR and SR/CSR. *** $p < 0.01$ in a one-way ANOVA ($n = 4$). Different letters indicate significant differences between ecological strategies groups (Tukey's test; $p < 0.05$).

## 4. Discussion

### 4.1. Dynamic Successional Patterns in Community Ecological Strategy Spectra

This study found Grime's CSR theory allowed the functional interpretation of tropical lowland rainforest communities along a successional gradient [15] as well as the identification of realized functional niches within the communities. Pierce et al. [1] found that the CSR strategies for species characteristic of primary succession from scree vegetation to siliceous alpine grassland, terminating with alpine *Nardus* pasture, were evident. Our work further

confirms that the applicability and effectiveness of the globally-corrected CSR ecological strategy spectrum approach was assessed from the perspective of regional succession [19].

This study found that the ecological strategy spectra varied significantly along the successional gradient. In all three successional stages, most species had intermediate values for components C and S. This is consistent with previous research results [45]. There are few R-selection ecological strategies, likely because our survey does not include herbaceous plants [41]. The proportion of species having each strategy type differed over time. Initially, most species had CS, S/CS or CS/CSR strategies, with only a few having C or CR/CSR strategies.

As succession proceeded, the proportion of species with these three dominant strategies changed, with CS and S/CS becoming less common (31.66% to 25.06% and 24.46% to 16.23%, respectively), while CS/CSR increased in frequency (13.42% to 21.81%). Only a few species possessed C or R/CR strategies. As forests matured (late successional stage), most species had CS/CSR, CS or S/CS strategies, while only a few species had C or R/CSR strategies. The diversity of ecological strategies indicates that plants take various trade-offs to make use of the acquired environmental resources [5,7,26].

However, the environmental driving force behind the changes of these strategies remains unclear. Strengthening these studies will contribute to reveal the relationship between environment and ecological strategies [41,46]. Use of this approach may help to predict patterns of species' functional trade-offs in a specific environment in addition to how community processes respond to the environment [1,41]. The number of strategies tended to decrease over the course of succession. Strategy richness was also negatively correlated with species richness. This is an interesting discovery. More species are thought to have more ecological strategy classifications [47].

However, the late succession stage with the largest number of species in this study showed the least ecological strategy classifications. This may be because, later in succession, after a long period of environmental screening [48], species' ecological strategies have converged [49,50]. The stability of functions shifts during the process of community assembly in secondary (restored) forests [47]. Therefore, although the number of species increases, the number of ecological strategies decreases. The differences in strategy richness (among community types) may be due to variation in the driving forces underlying successional processes, which may be affected by the interaction of multiple environmental factors [46,51].

*4.2. Community-Level Differences in Ecological Strategy Composition with Succession*

By summarizing the ecological strategy spectra for all succession stages, the CS strategy was found to be the most common strategy across all three successional stages. However, the proportion of species having a CS strategy (secondary strategy) was highest in the early stages of succession and was replaced by the CS/CSR strategy (tertiary strategy) in mature forests (30.40% for CS/CSR vs. 26.19% for CS). For all three successional stages, the proportion of species increased from primary to secondary to tertiary strategy.

Therefore, more species presented with complex versus simple functional trade-offs in the study communities. This is consistent with the research results of tropical forest ecological strategies in these four climatic zones [52]. Primary and secondary strategies were better represented early in succession, while tertiary strategies were more common during the mid-to-late stages of succession. This suggests that, over the course of succession, more complicated trade-offs among traits emerge. As a result, complex combinations of stress tolerance, competitive acquisition and resource allocation traits become more common.

Scholars have found that, if the influence of the plant community species composition and diversity on the community is ignored, the functional traits of individual plants cannot accurately reflect the ecosystem function [5,53]. Instead, the functional traits of plants at the community level can better reflect the ecosystem function [5]. Differences in CWM-CSR strategies may reflect changes in the community functional trait composition caused by

succession. The most competitive community (i.e., the highest C-value) was that of the mature forest, while the recently abandoned farmland had the lowest C-value.

The latter community was also the most stress-tolerant (i.e., the highest S-score), in contrast to the mature forest (the lowest degree of S-selection). This contradicts our hypothesis. The mid-successional stage community showed intermediate C- and S-values compared to the early and late stages. Stress tolerance (as S-values) was the highest early during succession and declined over time, likely as the disturbances associated with slash-and-burn cultivation slowly diminished with the passage of time [54]. During recovery from slash-and-burn cultivation, a positive feedback loop also develop [55], whereby vegetation growth provides richer resources to the environment [56], which then, in turn, further promotes plant growth.

As a result, the C-values (i.e., the ability to compete for resources) rise as resources become more abundant [48]. Therefore, when compared to the disturbance of slash-and-burn cultivation, the resource limitations occurring later in succession do not significantly limit community restoration. These resource limitations also play a positive role in the process of secondary succession after slash-and-burn cultivation—for instance, the litter improves the soil environment [57]. In secondary succession, forest habitats and ecosystem functions are restored.

## 5. Conclusions

This study demonstrated that the CSR framework (based on functional traits) is an effective method to evaluate how succession impacts the functional composition of forests. Over the course of succession in tropical lowland rainforests, both the ecological strategy spectra and CWM-CSR strategies were found to shift. Species with more complex balance and combinations of functional traits have greater survival advantages. This study expands the understanding of ecosystem function in successional forests and provides a perspective on research regarding the ecological strategy of the community building process.

**Author Contributions:** Conceptualization, X.L. and R.Z.; methodology, X.L. and R.Z.; formal analysis, C.C. and Y.W.; investigation, C.C., Y.W. and T.J.; writing—original draft preparation, C.C.; writing—review and editing, X.L. and H.Z.; visualization, C.C. and Y.W.; project administration, X.L.; funding acquisition, X.L. All authors have read and agreed to the published version of the manuscript.

**Funding:** This research was funded by the National Natural Science Foundation of China 31901210 and by the Doctoral Research Foundation of Liaocheng University 318051822.

**Acknowledgments:** We are grateful to the many people who contributed to this study, particularly Xiusen Yang and Yuecai Tang at the Bawangling Nature Reserve for their assistance in specimen identification and work in the field investigation. We would like to thank Emily Drummond at the University of British Columbia for her English editing of the manuscript.

**Conflicts of Interest:** The authors declare that they have no known competing financial interests or personal relationships that could have appeared to influence the work reported in this paper.

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
