# Peer review of "Ecological Strategy Spectra for Communities of Different Successional Stages in the Tropical Lowland Rainforest of Hainan Island"

_forests, doi:10.3390/f13070973_

Round 1
Reviewer 1 Report
The paper is of high quality. It deserves to be published. Just some minor corrections are needed there. Please, look to my comments below.
The Introduction is well written, but it is a bit long. I would recommend deleting the paragraph in lines 62-69; otherwise, it can be moved to the Material and Methods.
Line 96: “previous studies”. Please, add references to these studies. Now it is unclear what studies are meant there.
The text in lines 102-106 is not aim and tasks. This is the text for Materials and Methods, since it describes the sampling and methods used in this study. After moving this text to the Material and Methods, please, state clearly the main aim and research tasks of your study.
In Materials and Methods, some text fragments are marked by colours. Why?
Other parts of this section are well written, just add references to the R software and to certain packages used in the analysis each time, when they are mentioned. Please, use the format of MDPI journals for this purpose (it is not so in line 171).
The section Results is well written and presented. I have no serious suggestions. Some inconsistencies and difficult for understanding sentences are present, but I think that the authors will double-check the text during the revision and correct the text.
In Discussion, I suggest to avoid presenting the fragments with the obtained results (for instance, Fig. 5 and its description). It should be moved to the Results.
In addition, I recommend to use more references to compare the obtained results with other studies published previously. For instance, I mean the paragraph in lines 279-292. The changes in plant composition during the succession is well known and widely studied issues. Your results obtained by this method can (should) be supported by references where the same (or similar) results were obtained by other methods in several parts of the world. Add these references. I recommend to add references in other parts of the Discussion to highlight the international relevance of your results, as well as international applicability of this method in other regions of the world.
Lines 322-327: this paragraph should be a part of Conclusions. This is not a discussion of the obtained results.
Also, I recommend to pay more attention to the discussion of the results. In the section Results, the authors have presented a number of interesting results, which deserve discussing in light of the recent literature. If the authors connect their results with results of other studies obtained using the same or other methods, it will improve the final version of the paper.
Please, make the section Conclusions to be more related to the obtained results. Now, it is written by general words and conclusions.
Author Response
Dear editor and reviewers:
Thank you for your constructive comments and suggestions. These comments are very helpful for improving this manuscript. We have done our best to revise this manuscript according to your suggestions, and we hope that this revised version will meet the requirements for publication in forests.
Comments from reviewer # 1:
The Introduction is well written, but it is a bit long. I would recommend deleting the paragraph in lines 62-69; otherwise, it can be moved to the Material and Methods.
Response: Thank you very much for your suggestion. We adjusted the introduction according to your comments, and moved the sentences of lines 62-69 to the Material and Methods section.
Line 96: “previous studies”. Please, add references to these studies. Now it is unclear what studies are meant there.
Response: Thanks for your checks. We have added references here.
The text in lines 102-106 is not aim and tasks. This is the text for Materials and Methods, since it describes the sampling and methods used in this study. After moving this text to the Material and Methods, please, state clearly the main aim and research tasks of your study.
Response: Thank you for your constructive comments. We revised the texts and clearly explained the main purpose and task of this research. “In this study, forest ecological strategy spectra belonging to different successional stages were determined. Then the following two questions were discussed: (1) whether the forest ecological strategy spectra change with succession? (2) what is the effect of succession on forest ecological strategy composition?”
In Materials and Methods, some text fragments are marked by colours. Why?
Response: Thanks for your checks. This is a writing mistake, and we have corrected it.
Other parts of this section are well written, just add references to the R software and to certain packages used in the analysis each time, when they are mentioned. Please, use the format of MDPI journals for this purpose (it is not so in line 171).
Response: Thank you for your suggestion. We added the reference to R package and corrected it to the citation format of "MDPI" journal.
The section Results is well written and presented. I have no serious suggestions. Some inconsistencies and difficult for understanding sentences are present, but I think that the authors will double-check the text during the revision and correct the text.
Response: Thank you for your careful review of our manuscript. We revised the incorrect sentences.
In Discussion, I suggest to avoid presenting the fragments with the obtained results (for instance, Fig. 5 and its description). It should be moved to the Results.
Response: Thanks for your guidance, we have moved some research results from the discussion section to the results section.
In addition, I recommend to use more references to compare the obtained results with other studies published previously. For instance, I mean the paragraph in lines 279-292. The changes in plant composition during the succession is well known and widely studied issues. Your results obtained by this method can (should) be supported by references where the same (or similar) results were obtained by other methods in several parts of the world. Add these references. I recommend to add references in other parts of the Discussion to highlight the international relevance of your results, as well as international applicability of this method in other regions of the world.
Response: Thank you for your constructive comment. We discussed our results by combining the research of other authors in similar fields.
Lines 322-327: this paragraph should be a part of Conclusions. This is not a discussion of the obtained results.
Response: Thank you for your comment. We have moved this part to the conclusion section.
Also, I recommend to pay more attention to the discussion of the results. In the section Results, the authors have presented a number of interesting results, which deserve discussing in light of the recent literature. If the authors connect their results with results of other studies obtained using the same or other methods, it will improve the final version of the paper.
Response: Thank you very much for your constructive suggestion. We further discussed the reasons for the interesting results of this study, and the conclusions supported by our results.
Please, make the section Conclusions to be more related to the obtained results. Now, it is written by general words and conclusions.
Response: Thank you for your careful review of our manuscript. According to the results of this study and the discussion of the results, we revised the conclusion of this paper.

Reviewer 2 Report
Dear Authors,
Please find my comments in the attached file
Regards

Author Response
Dear editor and reviewers:
Thank you for your constructive comments and suggestions. These comments are very helpful for improving this manuscript. We have done our best to revise this manuscript according to your suggestions, and we hope that this revised version will meet the requirements for publication in forests.
Comments from reviewer # 2:
- You can not put abbreviations without explanation.
Response: Thanks for your suggestion. We have corrected our mistakes.
2.Your introduction part does not provide state of the art of the research question. It is not clear the present research state. Thus, the problem of the research is not formulated properly. Please improve your introduction part that the state of the research questions would be clearly presented. Then formulate the research problem.
Response: Thank you for your constructive comment. We summarized the research status of "ecological strategy", and put forward the problem that the change of ecological strategy with succession in tropical lowland rain forest is still unclear.
3.Please clarify the aim and the tasks of your study. Only hypothesis it is quite simple for this to represent clearly.
Response: Thank you for your suggestion. According to your suggestion, we pointed out the purpose and task of this article in the text. “In this study, forests ecological strategy spectra belonging to different successional stages were determined. Then the following two questions were discussed: (1) whether the forest ecological strategy spectra change with succession? and (2) what is the effect of succession on forest ecological strategy composition?”
4.Here you write about four communities, but in table 1 only three forest types are analyzed. Also, in all your work there is no analysis regarding slash and burn farmland? So how to understand this. Further you only analyze three successional stages, but how
Response: Thank you for pointing out the mistake. We have studied the ecological strategy spectrum of three successive stages. We have corrected the mistakes. Thank you again for checking. We added the analysis of slash-and-burn farmland in the section of materials, methods and discussion.
5.Analyze three successional stages, but how they are related to four communities of different successional stages is not clear.
Response: Thanks for your checks. This is our mistake in writing. We have revised four successional stages into three successional stages.
6.Here reference is a must
Response: Thank you very much for your checks. We have added references.
7.What was the method to decide where to establish the plot? How the place was selected?
Response: Thank you very much for your careful readings. we added “Abandoned slash-and-burn farmland, naturally-restored secondary forest, and a few undisturbed old-growth forests are distributed here. Information on the history of land-use for the plots was obtained from the management records of the Bawangling National Nature Reserve.”
8.What was done separately in these subplots that it is important to clarify the boundaries of them. What was the differences inside them? Othervice there was no need to make them
Response: We divided the plot of 1 hectare into 25 subplots of 20m*20m for the convenience of community investigation. Prevent omission or repetition of investigation due to excessive area.
9.You made no analysis regarding this data so why to mention this?
Response: According to your suggestion, we deleted the writing of "relative spatial coordinates" which was not involved in the text.
10.There were any sampling differences? how the plants were selected for the analysis in side the plot if them there more than 10?
Response: Random sampling is conducted if there are more than ten individuals in each species.
11.Please think how to reduce the number of abbreviations. It is very difficult to follow your work then you use so much of them.
Response: Thank you for your careful review of our manuscript. According to your opinion, we reduced the use of abbreviations.
12.Please clarify the methods used here in more detailed. It is not enough joust to write the name the package used. The procedure itself has to be clarified at least a bit.
Response: Thank you for your comments. We have further explained the data analysis section, citing the references of relevant R packages.
13.How does this data relates to the farmlands?
Response: This data is the number of species in four plots in each succession stage.
14.I think you did not clarify the methods how to estimate these strategies.
15.Nothing in methods part. How did you get them?
Response: Thank you for your suggestion. In the data analysis section of materials and methods, we added their introduction.
16.Is it so? If so, this neglects your work.
Response: Thank you for your checks. According to the research results, we correct it as follows “The number of strategies tended to decrease over the course of succession.”
17.It is not clear. There no differences between communities, but somehow differences between succession stages appeared. Like a wrote it is not clear the relation between the communities and succession stages.
Response: Thank you for your comments. We have made modifications, and there are significant differences among different succession stages.
18.I can see the successional stages. But I can’t see the communities involved here. Where is a sludged farm here?
Response: In each succession stage, we selected four plots of 1 hectare as communities. The dots in Figure 3 represent communities.
19.I think in this paragraph the comparison of your results with other authors results is lacking.
20.Please discuss your findings with other author findings. Also what is the meaning of your results in a broader scale?
Response: Thank you for your constructive guidance. We compared the research results with those published by other authors, found consistent and inconsistent views, and discussed them.
21.There was no differences between the communities you wrote in page 5 above figure 1.
Response: We modified the wrong expression in Figure 1, and the result shows that there are significant differences in ecological strategy spectrum in different succession stages.
22.Please clarify in more general form what now we know after your research done what we did not know before in the broader scale?
Response: Thank you for your opinion. We revised the conclusion to clarify our research results in a more general form.

Round 2
Reviewer 2 Report
Dear authors,
Thank you for taking my comments into account.
Regards